# Interpretable Ensemble-based Deep Learning Approach for Automated Detection of Macular Telangiectasia Type 2 by Optical Coherence Tomography

**Shahrzad Gholami** [1]   **Lea Scheppke** [2]   **Rahul Dodhia** [1]   **Juan Lavista** [1]   **Aaron Lee** [3][4]

## Abstract

We present an ensemble-based approach using deep learning models for the accurate and interpretable detection of Macular Telangiectasia Type 2 (MacTel) from a large dataset of Optical Coherence Tomography (OCT) scans. Leveraging data from the MacTel Project by the Lowy Medical Research Institute and the University of Washington, our dataset consists of 5200 OCT scans from 780 MacTel patients and 1820 non-MacTel patients. Employing ResNet18 and ResNet50 architectures as supervised learning models along with the AdaBoost algorithm, we predict the presence of MacTel in patients and reflect on interpretability based on the Grad-CAM technique to identify critical regions in OCT images influencing the models' predictions. We propose building weak learners for the AdaBoost ensemble by not only varying the architecture but also varying amounts of labeled data available for training neural networks to improve the accuracy and interpretability. Our study contributes to interpretable machine learning in healthcare, showcasing the efficacy of ensemble techniques for accurate and interpretable detection of rare retinal diseases like MacTel.

## 1. Introduction

Interpretable machine learning plays a vital role in healthcare, enabling accurate and transparent predictions for informed decision-making. Optical coherence tomography (OCT) imaging has emerged as a valuable tool for diagnosing and monitoring retinal diseases (Lee et al., 2017; Ting et al., 2019). In this study, we focus on the classification of Macular Telangiectasia Type 2 (MacTel) using OCT images, aiming to provide early and precise detection of this neurodegenerative disease (Issa et al., 2013). MacTel is a rare disease and historically was often misdiagnosed, though diagnosis has improved with increased awareness of the disease. Conventionally, MacTel diagnosis relies on a multimodal image set and a clinician being familiar with the disease (Chew et al., 2023). As OCTs have become more widely available in the clinic (De Fauw et al., 2018), we aim to find out if deep learning models can accurately predict MacTel based solely on the OCT image. Furthermore, information learned through these models will inform the research in this domain. Using deep learning framework for MacTel classification has been under-explored due to limited data availability (Loo et al., 2022). In this work, we propose an ensemble solution that enhances the interpretability and performance of MacTel classification models. Our methods employ deep learning models, such as ResNet18 and ResNet50, and leverage the AdaBoost ensemble method for model accuracy and interpretability improvement.

We conduct extensive experiments using a large OCT datasets and evaluate performance via standard metrics for classification tasks, including accuracy, precision, recall, F1 score, area under the curve (AUC), and average precision. Additionally, we utilize the Grad-CAM technique to visualize the regions of OCT images that contribute to the models' predictions, providing insights into the important features used for classification. Our experimental results demonstrate improvements in both interpretability and classification accuracy compared to using individual models. These findings highlight the potential of our proposed ensemble approach to accurately classify MacTel while providing meaningful explanations for the predictions.

## 2. Methods and Methodology

### 2.1. OCT Image Dataset

We use datasets obtained using a SPECTRALIS OCT device, collected through the MacTel Project Natural History Observation Registry Study, which includes 2636 OCT scans from 780 MacTel patients and 131 non-MacTel

[1]Microsoft AI for Good Research Lab [2]The Lowy Medical Research Institute [3]University of Washington [4]Roger and Angie Karalis Johnson Retina Center. Correspondence to: Shahrzad Gholami <sgholami@microsoft.com>.

*Workshop on Interpretable ML in Healthcare at International Conference on Machine Learning (ICML)*, Honolulu, Hawaii, USA. 2023. Copyright 2023 by the author(s).

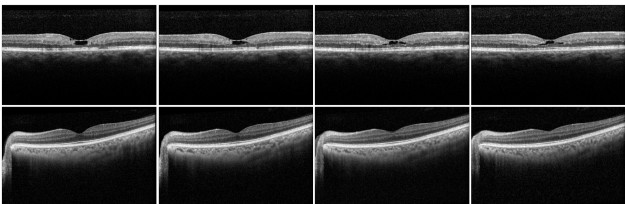

Figure 1. Example of several OCT slices for middle part of the scan for a patient with MacTel shown in the first row and a patient without MacTel shown in the second row.

patients, augmented with an additional 2564 scans of non-MacTel patients collected by the University of Washington (UW). The dimensions of the OCT volumes varied in terms of width and height. Figure 1 illustrates several OCT B-scans for a MacTel patient in the first row and a non-MacTel patient in the second row, focusing on the region around the fovea. To ensure uniformity in sample size, all volumes were resampled to a fixed dimension of 496 x 768 x 196 B-scans using linear interpolation. To streamline computations and concentrate on relevant areas, we select the central third of B-scans from each volume and resample them into three B-scans. These three B-scans were then combined to form an RGB image with three channels, where each channel represented a single B-scan. This flattening approach enabled us to leverage contextual information from neighboring B-scans in two-dimensional neural network architectures. The resulting dataset consisted of 5200 volumes. To enhance the model's robustness, we apply data augmentation techniques such as random horizontal flips and center crops. We randomly divide the dataset into training, validation, and test sets, with an 80:10:10 ratio at the patient level. The training and validation sets were used for model training and hyperparameter tuning, while the test set was reserved for the final model performance evaluation.

## 2.2. Deep Learning Models Training and Evaluation

We use supervised learning methods to train ResNet18 and ResNet50 architectures using the available labels to classify MacTel patients. We use PyTorch implementations for these models and stochastic gradient descent optimizer with a learning rate of 0.001 and a batch size of 32. To initialize the models' weights, we utilized pre-trained weights from ImageNet, a large-scale dataset consisting of natural images. Subsequently, we fine-tuned the models' weights on our OCT dataset for 100 epochs, incorporating early stopping based on the binary cross-entropy validation loss.

We explore the application of ensemble methods on OCT images to improve the performance of deep learning models. Previous studies explored ensemble techniques for patients with diabetic retinopathy based on fundus images (Jiang

et al., 2019) and based on OCT angiography images (Heisler et al., 2020), as well as for breast cancer detection (Zheng et al., 2020) and retinal vessel segmentation (Memari et al., 2017). To extend that idea, we propose building weak learners by not only varying the architecture but also leveraging varying amounts of labeled data available for training ResNet18 and ResNet50 architectures. We employ the AdaBoost algorithm to effectively combine the results of several individual models (Freund et al., 1999; 1996). The objective is to reduce the bias of each strong classifier and benefit from the diversity of the individual deep learning models within the ensembles.

To evaluate the performance of the trained models, we use several metrics, including accuracy, precision, recall, F1 score, AUC and average precision, which we compute on the reserved test set. We use Grad-CAM as the explainability method to further interpret the models and gain insight into their decision-making processes (Zhou et al., 2016). These methods allow us to visualize the regions of the OCT images that the models attended to when making predictions and provide a way to validate the models' predictions and gain insight into the underlying biological mechanisms (Altan, 2022; Tjoa & Guan, 2020; van der Velden et al., 2022; Saeed & Omlin, 2023).

## 2.3. Adaboost Algorithm

We propose the DL-AdaBoost algorithm, outlined in Algorithm 1, as a boosting approach to combine multiple hypothesis models (or weak learners), which are trained deep learning models, into a unified and improved model. The algorithm iteratively adjusts the weights assigned to each model based on their performance, aiming to minimize the weighted error. At each iteration, the sample distribution is updated to emphasize challenging samples. The algorithm terminates when the weighted error becomes convergent and reaches a stable state. The outputs of DL-AdaBoost are the learned weights along with the final integrated model, which is a weighted combination of the individual models. The weights assigned to each model, denoted as $\alpha$, reflect their contribution to the ensemble. To aggregate the predictions, we utilize $\alpha$ as weights to combine the predicted probabilities generated by the deep learning models. Additionally, we leverage the class activation maps (CAMs) inferred from the global pooling layer of the deep learning models (Selvaraju et al., 2017). The CAMs provide a visual representation, resembling a thermogram, highlighting suspected pathology within the image. We also combine CAMs of the weak learners based on the learned weights to captures the collective contributions of the individual models in the ensemble.

**Algorithm 1** DL-AdaBoost

---

**Input:** $(x_i, y_i)$ where $i = 1, ..., m$, $x_i$ is the feature vector, and $y_i$ is the label of $x_i$, hypothesis models $h_t$ for $t = 1, ..., T$, built based on a portion of labeled data and choice of architecture

**Initialize:** distribution of samples: $D_1(i) = \frac{1}{m}$

**repeat**

    **for** $t = 1$ **to** $T$ **do**

        Select hypothesis model $h_t$

        Compute weighted error $\epsilon = \mathbb{P}[h(x) \neq y]$

        Compute weight $\alpha = \frac{1}{2} \ln\left(\frac{1-\epsilon_t}{\epsilon_t}\right)$

        Update $D_{t+1}(i) = \frac{D_t(i)\exp(-\alpha y_i h_t(x_i))}{Z_t}$;

        where $Z_t$ is the normalization factor

    **end for**

**until** weighted error is convergent

**Output:** weights $\alpha$, final model $H(x) = \sum_{t=1}^{T} \alpha_t h_t(x)$

---

## 3. Results and Discussions

We show the test set results of our experiments in Table 1, showcasing the performance evaluation of ResNet50 and ResNet18 trained on different percentages of labeled data (i.e., weak learner). It is evident that both architectures exhibit satisfactory performance in MacTel classification. However, as the amount of labeled data increases, the performance of ResNet18 models consistently improves in comparison to ResNet50. This trend may be attributed to the higher susceptibility of ResNet50, which has approximately 23 million parameters, to overfitting, unlike ResNet18.

We construct three ensemble models using different combinations of weak learners, namely RESNET50-ADB, RESNET18-ADB, and RESNET(50&18)-ADB. In RESNET50-ADB, only weak learners based on the ResNet50 architecture are utilized in DL-AdaBoost. Similarly, RESNET18-ADB consists of weak learners exclusively from the ResNet18 architecture. Finally, RESNET(50&18)-ADB incorporates weak learners from both ResNet18 and ResNet50 architectures. Table 2 presents the weights learned for each ensemble. These weights signify the importance assigned to each individual model within the ensemble.

Table 3 displays the performance results of these ensembles on the test set. The DL-AdaBoost ensemble method effectively enhances the overall model performance across almost all evaluated metrics. The ensembles consistently outperform the individual models trained with different percentages of labeled data, highlighting the advantages of incorporating multiple weak learners within an ensemble framework. Notably, RESNET(50&18)-ADB exhibits the highest performance among the ensembles. This can be attributed to its utilization of a larger number of

individual learners from both the ResNet50 and ResNet18 architectures. The combination of these architectures leverages their complementary strengths, resulting in improved performance compared to ensembles based on individual architectures alone.

Additionally, we conduct a visual analysis of the Grad-CAM results, as shown in Figure 2 and 3. In Figure 2, we present the results for individual and ensemble models when ResNet50 (Figure 2a) and ResNet18 (Figure 2b) architectures are used separately. Panels a-e display the results for individual models trained on various percentages of labeled data, while panels c and f show the flattened RGB version of the OCT image and the corresponding Grad-CAM based on the ensemble model, respectively.

In Figure 3, we present the Grad-CAM results when both architectures are used in the ensemble model. Panels a-d represent the results for individual models trained on different percentages of labeled data for ResNet50, while panels f-i display the results for individual models trained on different percentages of labeled data for ResNet18. Panel e and j show the flattened OCT image and the Grad-CAM based on the ensemble model RESNET(50&18)-ADB, respectively. By comparing the results of the ensemble model RESNET(50&18)-ADB to the individual weak learners, we observe that the ensemble approach yields more focused results on the pathology of interest. This indicates the effectiveness of the ensemble approach in enhancing the localization of the target pathology, further supporting the advantage of leveraging multiple models in the ensemble framework.

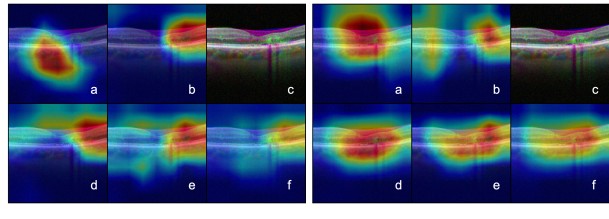

(a) Results for ResNet50      (b) Results for ResNet18

*Figure 2.* Grad-CAM results for individual and ensemble models for a patient with MacTel when architectures are used separately; a-e show results for individual models learned based on various amounts of labeled data from 10% to 100%, c and f show flattened OCT image and grad-CAM based on ensemble model, respectively.

## 4. Conclusion

Our approach for MacTel classification using OCT images, employing interpretable deep learning models and ensemble methods, demonstrated promising results. The ResNet18 and ResNet50 architectures effectively classified MacTel, and the ensemble method further improved overall model performance. The visual analysis

*Table 1.* Test set performance of individual models trained on various amounts of labeled data for ResNet50 and ResNet18

| ARCHITECTURE | % OF LABELS | AUC | AVG PREC | ACC | F1 | RECALL | PRECISION |
|---|---|---|---|---|---|---|---|
| RESNET50 | 10% | 0.946 | 0.939 | 0.884 | 0.868 | 0.844 | 0.892 |
| | 25% | 0.958 | 0.953 | 0.898 | 0.884 | 0.867 | 0.903 |
| | 50% | 0.937 | 0.934 | 0.870 | 0.861 | 0.893 | 0.831 |
| | 100% | 0.947 | 0.935 | 0.898 | 0.888 | 0.898 | 0.878 |
| RESNET18 | 10% | 0.855 | 0.842 | 0.764 | 0.762 | 0.840 | 0.697 |
| | 25% | 0.943 | 0.926 | 0.858 | 0.828 | 0.760 | 0.910 |
| | 50% | 0.949 | 0.946 | 0.868 | 0.864 | 0.929 | 0.807 |
| | 100% | 0.965 | 0.963 | 0.890 | 0.878 | 0.876 | 0.879 |

*Table 2.* Weights learned by DL-AdaBoost for each individual model used in the ensemble: RESNET50-ADB: ensemble only based on ResNet50, RESNET18-ADB: ensemble only based on ResNet18, and RESNET(50&18)-ADB: ensemble based on both architectures

| ARCHITECTURE | RESNET50 | | | | RESNET18 | | | |
|---|---|---|---|---|---|---|---|---|
| % OF LABELS | 10% | 25% | 50% | 100% | 10% | 25% | 50% | 100% |
| RESNET50-ADB | 0.231 | 0.274 | 0.210 | 0.285 | - | - | - | - |
| RESNET18-ADB | - | - | - | - | 0.170 | 0.224 | 0.275 | 0.332 |
| RESNET(50&18)-ADB | 0.130 | 0.158 | 0.110 | 0.149 | 0.072 | 0.096 | 0.130 | 0.155 |

*Table 3.* Test set performance of ensembles; RESNET50-ADB: ensemble only based on ResNet50 architecture, RESNET18-ADB: ensemble only based on ResNet18 architecture, and RESNET(50&18)-ADB: ensemble based on both architectures

| ENSEMBLE MODEL | AUC | AVG PREC | ACC | F1 | RECALL | PRECISION |
|---|---|---|---|---|---|---|
| RESNET50-ADB | 0.965 | 0.964 | 0.916 | 0.906 | 0.902 | 0.910 |
| RESNET18-ADB | 0.968 | 0.960 | 0.918 | 0.909 | 0.907 | 0.911 |
| RESNET(50&18)-ADB | 0.975 | 0.973 | 0.922 | 0.912 | 0.902 | 0.923 |

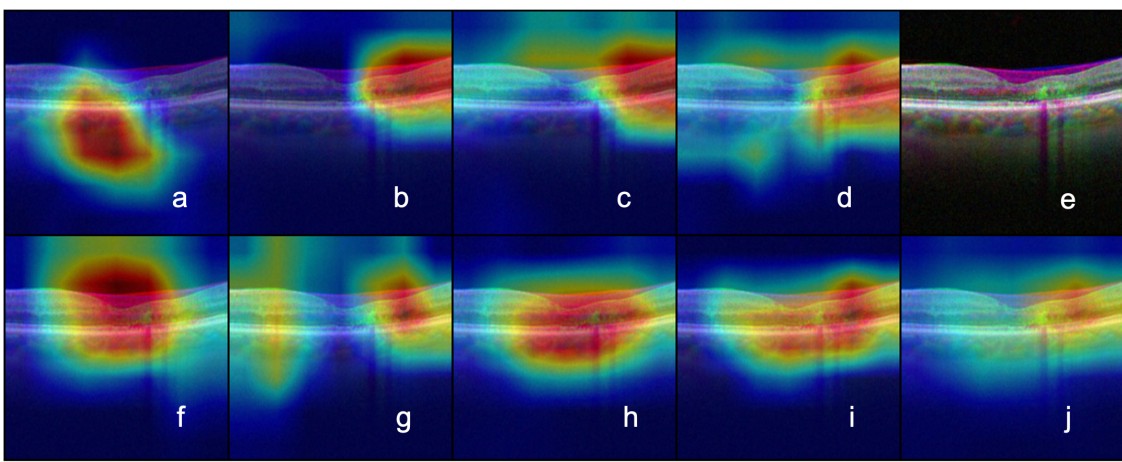

*Figure 3.* Grad-CAM results for individual and ensemble models for a patient with MacTel when both architectures are used in the ensemble; a-d show results for individual models learned based on various amounts of labeled data from 10% to 100% for ResNet50, f-i show results for individual models learned based on various amounts of labeled data from 10% to 100% for ResNet18, e and j show flattened OCT image and grad-CAM based on ensemble model, respectively.

of Grad-CAM results highlighted the ensemble's ability to provide focused and localized predictions. These findings contribute to interpretable machine learning in healthcare, enhancing the interpretability and performance of MacTel classification. Future research can explore the approach's applicability to other retinal diseases and incorporate additional interpretable machine learning techniques for further improvement in interpretability and clinical utility.

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
