# OpenReview forum: "Interpretable Ensemble-based Deep Learning Approach for Automated Detection of Macular Telangiectasia Type 2 by Optical Coherence Tomography"
_ICML.cc/2023/Workshop/IMLH — IMLH 2023 PosterShortPaper_

### Official Review · Reviewer_q1Lo · 2023-06-10
**Comments on IMLH23, Submission 69**

**Rating:** 6
**Confidence:** 5

**Review:**

The author has presented an ensemble-based deep learning model aimed at accurately and interpretable detection of Macular Telangiectasia Type 2 (MacTel) using a large dataset of Optical Coherence Tomography (OCT) scans. AdaBoost algorithm is employed to ensemble the single model and enhance the classification performance, while the Grad-CAM technique is utilized to identify crucial regions in OCT images that influence the models' predictions. The results demonstrate the effectiveness of the proposed model, and the paper is easily comprehensible.

However, there are a few issues that need to be addressed in this paper. Firstly, it is necessary to emphasize the authors' contribution, as the method appears to be lacking novelty. Secondly, clarification is needed regarding whether the dataset was divided at the patient level or image level. To prevent cross-contamination between the training and test datasets, it is advisable to divide the dataset at the patient level.

---

### Official Review · Reviewer_Gm1K · 2023-06-12
**Interpretable Ensemble-based Deep Learning Approach for Automated Detection of Macular Telangiectasia Type 2 by Optical Coherence Tomography**

**Rating:** 5
**Confidence:** 4

**Review:**

**Summary:** The author propose to use ensembles ResNet with adaboost to detect the Macular Telangiectasia Type 2 from optical coherence tomography. Furthermore, they use GradCAM to interpret the trained model. Their result shows that the proposed ensemble method outperforms individual models on model performance and interpretability.

**Strength:**
1. The paper aims to perform automated Macular Telangiectasia Type 2 detection, which is an underexplored yet important area. The work shed light on the future works with the collected dataset.
2. The author empirically shows the improvement of using the ensemble method on the accuracy and interpretability.

**Weakness:**
1. The author did not describe the reason of selecting ResNet. As in a related work (chrome-extension://efaidnbmnnnibpcajpcglclefindmkaj/https://www.ncbi.nlm.nih.gov/pmc/articles/PMC8365558/pdf/nihms-1727321.pdf), they used CNN.
2. Although it is a new application, the novelty is relatively low since it’s more like an application work.
3. Some parts in the paper is hard to follow, for example, what is D in Algorithm 1?
4. Please correct some typos, for example, leaner in line 105 should be learner.

---

### Meta-Review · Area_Chair_4FG5 · 2023-06-20

**Recommendation:** Accept (Poster)
**Confidence:** 5

**Metareview:**

This paper received two borderline ratings from reviewers. Though there are several concerns, the author evaluated a novel interpretable method for a crucial application. The study has a potential impact on the clinical community. As a short paper, the authors performed concise qualitative and quantitative metrics. I suggest the author can further improve the manuscript with the reviewer's comments.

---

### Decision · Program_Chairs · 2023-06-20

Accept (Poster Short Paper)